# A Physical Phenomenon for the Fractional Nonlinear Mixed Integro-Differential Equation Using a General Discontinuous Kernel

**Sharifah E. Alhazmi** [1,*] and **Mohamed A. Abdou** [2]

1   Mathematics Department, Al-Qunfudah University College, Umm Al-Qura University, Al-Qunfudhah 28821, Saudi Arabia
2   Department of Mathematics, Faculty of Education, Alexandria University, Alexandria P.O. Box 21526, Egypt
*   Correspondence: sehazmi@uqu.edu.sa

**Abstract:** In this study, a fractional nonlinear mixed integro-differential equation (Fr-NMIDE) is presented and has a general discontinuous kernel based on position and time space. Conditions of the existence and uniqueness of the solution is provided through the principal form of the integral equation, based on the Banach fixed point theorem. After applying the properties of a fractional integral, the Fr-NMIDE conformed to the Volterra–Hammerstein integral equation (V-HIE) of the second kind, with a general discontinuous kernel in position with the Hammerstein integral term and a continuous kernel in time to the Volterra term. Then, using a technique of the separating method, we obtained HIE, where its physical coefficients were variable in time. The Toeplitz matrix method (TMM) and its schemes were used to obtain a nonlinear algebraic system by studying the convergence of the system. The Maple 18 program was implemented to present the numerical results, along with corresponding errors.

**Keywords:** fractional; integro-differential equation; Volterra–Hammerstein; discontinuous kernel; Toeplitz matrix method





## 1. Introduction

As integral equations, integro-differential equations and fractional integro-differential equations (IEs/IDEs/fIDEs) can be used to simulate a wide range of problems in the basic sciences, many scientists have focused their attention on presenting the solutions for these systems. These equations have played a significant role in finding solutions using diverse methods, which is in line with the rapid development in finding the answers to diverse problems originating from the basic sciences. Currently, several studies have concentrated on creating more sophisticated and effective techniques for solving the IEs/IDEs, such as the Rieman–Stieltjes integral conditions [1,2], Lerch polynomials method [3] and Legendre–Chebyshev spectral method [4], along with the numerical observations based on semi-analytical approaches, e.g., Adomian's decomposition method [5] and HOBW method [6]. The linear/nonlinear equations (IEs/IDEs/ fIDEs) have various uses in fluid mechanics [7], Stokes flow [8], airfoil [9], quantum mechanics [10], integral models [11], mathematical engineering [12], nuclear physics [13] and the theory of laser [14]. The orthogonal polynomials method is considered one of the most significant operators used to solve various scientific problems. Alhazmi [15] used a new technique based on the separation of variables and the orthogonal polynomials method to obtain many spectral relationships based on the mixed integral equation using a generalized potential kernel. Nemati et al. [16] applied the Legendre polynomials scheme for the outcomes of a second-order two-dimensional (2D) Volterra integral model, together with a continuous kernel. Mirzaee and Samadyar [17] discussed the convergence of 2D-orthonormal Bernstein collocation method for solving 2D-mixed Volterra-Fredholm integral equations. Basseem

and Alalyani [18] used Chebyshev polynomials to discuss the numerical solution of the quadratic integral equation with a logarithmic kernel. Katani [19] implemented a quadrature scheme for the numerical outcomes of the second kind of Fredholm integral model. Al-Bugami [20] used the Simpson and Trapezoidal schemes to perform numerical representations based on an integral model using 2D surface crack layers. Brezinski and Zalglia [21] used the extrapolation approach to achieve numerical computing results based on the second kind of nonlinear integral model that has a continuous kernel. Baksheesh [22] proposed using the Galerkin scheme to find the approximate results based on the Volterra integral equations of the second kind, which have discontinuous kernels. Alkan and Hatipoglu [23] applied the sinc-collocation method for solving the Volterra-Fredholm IDEs of fractional order. Mosa et al. [24] studied the semi-group scheme to assess uniqueness and existence based on the partial and fractional integro models of heat performance in the Banach space using the Adomian decomposition scheme. Bin Jebreen and Dassios [25] proposed an efficient algorithm to find an approximate solution via the wavelet collocation method for fractional Fredholm integro-differential equations. Akram et al. [26] interpreted the collocation approach to tackle the fractional partial integro-differential equation by employing the extended cubic B-spline Abdelkawy et al. [27] applied the Jacobi–Gauss collocation method after using the Riemann–Liouville fractional integral and derivative fractional to obtain the approximate solution for variable-order fractional integro-differential equations with a weakly singular kernel.

The initial value of the Fr-NMIDE is presented as:

$$\mu \frac{\partial^\alpha \Phi(x,t)}{\partial t^\alpha} + v\Phi(x,t) = \lambda \int_\Omega k(|x-y|)\Phi^m(y,t)dy + g(x,t), (\Phi(x,0) = \psi(x)). \quad (1)$$

Here, $g(x,t)$ and $\Phi(x,t)$ are the known and unknown continuous functions, respectively, in $L_2[\Omega]XC[0,T]$. $\Omega$ is the integration domain and $m = 1, 2, \ldots M$. In addition, $\mu$ indicates the constant values of Equation (1) and $\lambda$ and $v$ are constants and have physical sense. The kernel $k(|x-y|)$, in general, has a singular term. For reference, the essential properties and definitions have been stated using fractional calculus theory.

**Definition 1.** *For the function* $f : (0, \infty) \to \mathbb{R}$*, the fractional Riemann–Liouville integral with order* $\alpha > 0$ *is shown as [28]:*

$$I_{0^+}^\alpha l(t) = \frac{1}{\Gamma(\alpha)} \int_0^t (t-s)^{\alpha-1} l(s)ds.$$

*In addition, we define the Caputo derivatives of order as* $\alpha > 0$

$$D_{0^+}^\alpha l(t) = \frac{1}{\Gamma(n-\alpha)} \int_0^t \frac{l^{(n)}(s)ds}{(t-s)^{\alpha-n+1}}, (n < \alpha \le n+1).$$

The time Abel kernel $(t-\tau)^{\alpha-1}, \forall t, \tau \in [0,T], 0 \le \tau \le t \le T < 1$ satisfies the following features: $0 \le t_1 \le t_2 \le t \le T < 1$ for every continuous function $h(t)$. Integrals

$$\int_0^t (t-\tau)^{\alpha-1} h(\tau)d\tau, \max_{0 \le t \le T} \int_0^t (t-\tau)^{\alpha-1}d\tau, \int_{t_1}^{t_2} (t-\tau)^{\alpha-1} h(\tau)d\tau,$$

are the continuous time function, i.e.,

$$\left| \int_0^t (t-\tau)^{\alpha-1} h(\tau)d\tau \right| \le M.$$

In this paper, Section 2 presents the fractional definition to obtain a NMIE. Then, the theorem-based Banach fixed point is discussed to prove the existence and uniqueness of the solution of NMIE. Section 3 presents the convergence of the solution. Section 4 indicates the technique of separation of the variables for the Hammerstein integral model

in position and its coefficients. This scheme would help researchers choose the time known function in an easier way, enabling them to choose the necessary time to obtain the required results. Section 5 indicates the convergence analyses of the Hammerstein integral equation. Section 6 shows the Toeplitz matrix scheme to conform the Hammerstein integral model of the nonlinear algebraic system (NAS). The TMM is considered the best numerical method for solving singular integral equations, where the singular terms disappear and we have simple integrals. Section 7 represents the NAS convergence. Section 8 provides the convergence of the error using one of the famous theorems. Section 9 provides the numerical solutions through Maple 18, together with the kernel of the nonlinear integral equation that takes the logarithmic form, the Carleman function and Hilbert kernel. In addition, the corresponding errors are computed.

## 2. The Solution's Existence and Uniqueness

The fundamental Caputo fractional integral is used to find the second order NMIE as:

$$\mu\Phi(x,t) + \frac{v}{\Gamma(\alpha)}\int_0^t (t-\tau)^{\alpha-1}\Phi(x,\tau)d\tau - \frac{\lambda}{\Gamma(\alpha)}\int_0^t \int_\Omega (t-\tau)^{\alpha-1}k(|x-y|)\Phi^m(y,\tau)dyd\tau = f(x,t),$$

$$f(x,t) = \frac{1}{\Gamma(\alpha)}\int_0^t (t-\tau)^{\alpha-1}g(x,\tau)d\tau + \psi(x), \qquad 0 < \alpha < 1.$$

(2)

In Equation (2), the free term $f(x,t) \in L_2(\Omega) \times C[0,T]$ and the unknown function $\Phi(x,t)$ will be discussed in the same space, $L_2(\Omega) \times C[0,T]$, along with the discontinuous kernel $k|x-y| \in L_2([\Omega] \times [\Omega])$. The discontinuous kernel of time $(t-s)^{\alpha-1}, 0 < \alpha < 1$, is considered in class $C[0,T]$, where $T < 1$.

To prove the existence of NMIE (2) based on the unique results, the integral operator form is given below:

$$\bar{U}\Phi(x,t) = \frac{\lambda}{\mu}U_2\Phi(x,t) - \frac{v}{\mu}U_1\Phi(x,t) + \frac{1}{\mu}f(x,t),$$

$$U_1\Phi(x,t) = \frac{1}{\Gamma(\alpha)}\int_0^t (t-\tau)^{\alpha-1}\Phi(x,y,\tau)d\tau,$$

(3)

$$U_2\Phi(x,t) = \frac{1}{\Gamma(\alpha)}\int_0^t \int_\Omega (t-\tau)^{-1+\alpha}k(|y-x|)\Phi(y,\tau)dyd\tau.$$

The following conditions are presented as:
(i) (i-a) The position kernel $k(|x-y|)$ satisfies

$$\left\{\int_\Omega \int_\Omega k(|x-y|)^2 dxdy\right\}^{\frac{1}{2}}d\tau = C, \quad (\text{ C—constant }).$$

(i-b) Therefore, the kernel of position and time $(t-\tau)^{\alpha-1}k(|x-y|)$ in $L_2(\Omega) \times C[0,T]$ satisfies

$$\max_{0 \le t \le T}\int_0^t \left\{\int_\Omega \int_\Omega \left|(t-\tau)^{\alpha-1}k(|x-y|)\right|^2 dxdy\right\}^{\frac{1}{2}}d\tau = \frac{T^\alpha C}{\alpha} \quad (0 < \alpha < 1).$$

(ii) The continuous function $f(x,t) \in L_2(\Omega) \times C[0,T]$ and its norm is shown as
$\|f(x,t)\|_{L_2(\Omega) \times C[0,T]} = \max_{0 \le t \le T}\left|\int_0^t \{\int_\Omega |f(x,\tau)|^2 dx\}^{\frac{1}{2}}d\tau\right| = G$ and $G$ is taken as a constant.
(iii) The decreasing function $Q > P$, constant $Q > Q_1$ and $\Phi^m(x,t)$ are used as:
(iii-a) $\max_{0 \le t \le T}\left|\int_0^t \{\int_\Omega |\Phi^m(x,\tau)|^2 dx\}^{\frac{1}{2}}d\tau\right| \le Q_1\|\Phi(x,t)\|_{L_2(\Omega) \times C[0,T]}.$
(iii-b) $\quad |\Phi_1{}^m(y,t) - \Phi_2{}^m(y,t)| \le N(t,x)|\Phi_1(x,t) - \Phi_2(x,t)|, |N(t,x)| = p.$

**Theorem 1** (Banach Fixed Point [29])**.** *Consider $X = (X, d)$ to be a metric space, where $X \neq \{\Phi\}, \Phi$ is a null set. Suppose $X$ is complete and $T : X \rightarrow X$ is the $X$ contraction, then $T$ contains exactly a single fixed point.*

To prove this, $K$ shows the contraction operator on $B$ with integral form $K\Psi = \Psi, \Psi$ presents the unique form of the solution, the general kernel $H(|y - x|, (t - \tau) = (t - \tau)^{\alpha-1} \times k(|y - x|)$ satisfies in (i-b), $L_2(\Omega) \times C[0, T]$ is the domain of integration, with respect to position $\Omega$ and the time $t, \tau \in [0, T], 0 \leq \tau \leq t \leq T < 1$.

**Theorem 2.** *Principal theorem: The NMIE (2) with the use of above conditions and the space $L_2(\Omega) \times C[0, T]$ takes the form of:*

$$|\mu|\Gamma(\alpha + 1) > (v + CQ\lambda)T^{\alpha}. \tag{4}$$

Lemmas (1) and (2) must be proven to satisfy the above theorem.

**Lemma 1.** *Under conditions (i) to (iii-a), the $\bar{W}$ operator maps the space $L_2(\Omega) \times C[0, T]$ onto itself:*

**Proof.** Equation (3) is used to prove

$$\|\bar{U}\Phi(x, t)\| \leq \frac{\lambda}{|\mu|}\|U_2\Phi(x, t)\| + \frac{1}{|\mu|}\|f(x, t)\| + \frac{|v|}{|\mu|}\|U_1\Phi(x, t)\|.$$

Using (i)–(iii-a) and the inequality of Cauchy–Schwarz, we have:

$$\|\bar{U}\Phi(x, t)\| \leq \frac{Q}{|\mu|} + \sigma\|\Phi(x, t)\|, \left(\sigma = \frac{(v + \lambda CQ)T^{\alpha}}{|\mu|\Gamma(\alpha + 1)}\right). \tag{5}$$

In the above statement, the $\bar{U}$ operator maps the $S_r$ ball as:

$$r = \frac{Gr(\alpha + 1)}{[|\mu|\Gamma(\alpha + 1) - (v + \lambda CQ)T^{\alpha}]}. \tag{6}$$

As $r > 0, G\Gamma(\alpha + 1) > 0$, therefore $\sigma < 1$. Furthermore, the inequality (5) includes the boundedness operators $U_1, U_2$ and $\bar{U}$. $\square$

**Lemma 2.** *If (i)-(iii-b) conditions are fulfilled, then $\bar{U}$ operator is a contractive-based Banach space $L_2(\Omega) \times C[0, T]$.*

**Proof.** For $\Phi_1(x, t)$ and $\Phi_2(x, t)$ functions using $L_2(\Omega) \times C[0, T]$ space, Formula (3) becomes:

$$\|\bar{U}(\Phi_1(x, t) - \Phi_2(x, t))\| \leq \frac{|v|}{|\mu|}(\|U_1(\Phi_1(x, t) - \Phi_2(x, t))\| + \|U_2(\Phi_1(x, t) - \Phi_2(x, t))\|),$$

The conditions (i), (ii) and (iii-b) have been applied to the Cauchy–Schwarz inequality as:

$$\|\bar{U}(\Phi_1(x, t) - \Phi_2(x, t))\| \leq \sigma\|\Phi_1(x, t) - \Phi_2(x, t)\|. \tag{7}$$

Inequality (7) presents the operator $\bar{U}$ (contraction operator), which shows the continuity in the $L_2(\Omega) \times C[0, T]$ space. $\square$

## 3. Convergence of the Solution

Consider the simple iteration $\{\Phi_1(x,y),\ldots,\Phi_{n-1}(x,t),\Phi_n(x,t),\ldots\} \subset \Phi(x,t)$, where the two functions $\{\Phi_{n-1}(x,t),\Phi_n(x,t)\}$ are used to satisfy

$$\mu(\Phi_n(x,t) - \Phi_{n-1}(x,t)) + \frac{v}{\Gamma(\alpha)}\int_0^t (t-\tau)^{\alpha-1}(\Phi_{n-1}(x,t) - \Phi_{n-2}(x,t))d\tau = \frac{\lambda}{\Gamma(\alpha)}\int_0^t\int_\Omega (t-\tau)^{\alpha-1}k(|x-y|)\big(\Phi_{n-1}^m(y,\tau) - \Phi_{n-2}^m(y,\tau)\big)dyd\tau. \tag{8}$$

Consider

$$\Phi_n(x;t) = \sum_{i=0}^n \Psi_i(x;t), \tag{9}$$

where

$$\Psi_n(x,t) = \Phi_n(x,t) - \Phi_{n-1}(x,t); (n \geq 1), \Psi_0(x,t) = f(x,t).$$

Equation (8) is updated by using Equation (9):

$$\mu\|\Psi_n(x,t)\| \leq \frac{v}{\Gamma(\alpha)}\left\|\int_0^t (t-\tau)^{\alpha-1}\Psi_{n-1}(x,t)d\tau\right\| + \frac{\lambda}{\Gamma(\alpha)}\left\|\int_0^t\int_\Omega (t-\tau)^{\alpha-1}k(|x-y|)\Psi_{n-1}^m(y,\tau)dyd\tau\right\|.$$

Taking $n = 1$, the above formula becomes:

$$\|\Psi_1(x,t)\| \leq \sigma G, \left(\sigma = \frac{(v+\lambda CQ)T^\alpha}{|\mu|\Gamma(\alpha+1)}\right),$$

and we see

$$\|\Psi_n(x,t)\| \leq \sigma^n G, \qquad \sigma < 1. \tag{10}$$

Equation (10) shows the convergent sequence $\{\Psi_n(x,t)\}$ uniformly. Moreover, it provides the convergent solution of the sequence $\{\Phi_n(x,t)\}$. As $\Psi_i(x;t)$ is continuous and $\Phi(x,t) = \lim_{n\to\infty}\Phi_n(x,t) = \lim_{n\to\infty}\sum_{i=0}^n\Psi_i(x;t)$, $\Phi(x;t)$ is uniformly continuous with an infinite $\{\Phi_n(x,t)\}_{n=0}^\infty$ series. This proves the lemma.

## 4. Separation of Variables Scheme

In the problems of mathematical physics, we find that researchers are interested in finding the unidentified potential function, which is linked to time and position. A variety of methods can be used to obtain the unknown function. One of these methods is time division that turns the mixed integral equation into an algebraic system of integral equations. Researchers apply the separating variable method to solve the mixed integral equation using the coefficients of the space functions. Moreover, these time coefficients are in the form of an integral operator of the Volterra type (Jan [30,31]). This scheme helps researchers to choose the time known function in an easier way, which enables them to choose the necessary time to obtain the required results. The unidentified and known functions $\Phi(x,t)$ and $g(x,t)$ are shown in the separation form as:

$$\Phi(x,t) = X(x)Y(t), \quad g(x,t) = b(x)Y(t), \qquad Y(0) \neq 0, \tag{11}$$

where $X(x)$ is an unknown function in a position that is to be determined, $b(x)$ is the given function in a position and $Y(t)$ shows the known function in time.

The time function is chosen in the form of a series based on the polynomial constants. This form helps the researcher to categorize the function of time based on the constants as a famous function or time representation in several other forms. It is noted that time series convergence is based on the premise of the experiment time being less than one and the start time is not equal to zero. Assume that

$$Y(t) = t^\alpha \sum_{n=0}^\infty a_n t^n, \qquad a_0 \neq 0, \qquad t \in [0,T], \qquad T < 1. \tag{12}$$

Using Equations (11) and (12) in Equation (2), shown as

$$\mu X(x) - \rho(t) \int_\Omega k(|y - x|) X^m(y) dy = H(x,t),\tag{13}$$

where

$$\begin{cases} \rho(t) = \lambda Q(t) \int_0^t (t - \tau)^{\alpha-1} Y^m(\tau) d\tau, m = 1, 2, \ldots, M, \\ Q(t) = \left[ v \int_0^t (t - \tau)^{\alpha-1} Y(\tau) d\tau + \mu \Gamma(\alpha) Y(t) \right]^{-1}, \\ H(x,t) = Q(t) \left[ \Gamma(\alpha) \psi(x) + h(x) \int_0^t (t - \tau)^{\alpha-1} Y(\tau) d\tau \right]. \end{cases}\tag{14}$$

By using the updated form of Equation (13) by using Equation (14), the reader can use the separation of time scheme for the nonlinear mixed integral model in time and position, which leads to a nonlinear integral equation in a position with coefficients linked to $t$, where $t \in [0,T], T < 1$. Moreover, Equation (13) represents that the single solution condition on the nonlinear integral equation is an equivalence relationship between position and time, which is given as:

$$\|k(|x - y|)\| \le \delta,$$

$$\delta = \left| \left[ \mu \Gamma(\alpha) Y(t) + v \int_0^t (t - \tau)^{\alpha-1} Y(\tau) d\tau \right] \left[ \lambda \int_0^t (t - \tau)^{\alpha-1} Y^m(\tau) d\tau \right]^{-1} \right| < 1.\tag{15}$$

Equation (15) shows the relationship between the position represented in the kernel form (which represents the properties of matter) and the time required for the continuity of these properties, which is known under the condition that there is a single solution. It is noted that at a certain time, an increase that exceeds the standard value of the nucleus may occur, and this leads to instability.

## 5. Convergence Investigations Based on Nonlinear Integral Model

To check the nonlinear integral model (13) convergence, the solution sequence takes the form:

$$X(x) = \{X_0(x), X_1(x), \ldots, X_m(x), \ldots X_\ell(x), \ldots\},$$

where $\psi_n$ and $\psi_m$ are two distinct arbitrary partial sums of sequence $\{X_j(x)\} : (n > m)$, and

$$d(\psi_n, \psi_\ell) = \max_{n,m \in j} |X_n - X_\ell| \le \delta d(X_n, X_{\ell-1}) \le \delta^2 d(X_n, X_{\ell-2}) \le \ldots \le \delta^\ell d(X_n, X_0),$$

where $\delta$ is defined in the Equation (15). Finally, we have

$$d(\psi_n, \psi_\ell) \le \frac{\delta^\ell}{1-\delta} d(\psi_1, \psi_0); \quad 0 < \delta < 1.\tag{16}$$

As $\ell \to \infty$ and for the fixed values, $d(\psi_1, \psi_0)$ must approach zero. Hence, it is concluded that $\{\psi_n\}$ shows a Cauchy sequence throughout the metric space, which shows the convergent series.

## 6. Toeplitz Matrix Method (Abdou et al. [32])

Many numerical methods have been used to solve integral equations with continuous or unconnected kernels. In singular integral equations, the best way to solve them is the Toeplitz matrix method (TMM), due to the following reasons: The singular term directly disappears, being transformed into simple integrals that can be solved quickly, and then, forms a linear/nonlinear algebraic system of equations. The degree of convergence in the relative error is less than in the other methods.

There are some methods that give a quick approximation when studying the error, including a circulated preconditioned iterative scheme. Xian et al. [33] considered the conjugate gradient technique with a circulated preconditioned scheme for a linear discretized model. They obtained a matrix whose coefficient kept the structure of a symmetric Toeplitz-plus-tridiagonal. They also studied linear large-scale systems with coefficient matrices based on the non-Hermition Toeplitz and applied a product of a fast Toeplitz matrix-vector through iterative solvers for a linear discretized model (circulant preconditioned method). This numerical scheme provides fast results to reduce computational costs, using $O(m^3)$ to $O(m \log m)$, and storage through $O(m^2)$ to $O(m)$ deprived of loss compression, where $m$ is the number of spatial grid nodes.

For TMM, consider $\Omega \in (-1, 1)$, using the nonlinear integral term of (2) as:

$$
\int_{-1}^{1} k(|y - x|) X^m(y) dy = \sum_{\ell=-N}^{N-1} \int_{a=\ell h}^{(\ell+1)h} k(|y - x|) X^m(y) dy
$$

$$
= \sum_{\ell=-N}^{N-1} [A_\ell(x) X^m(\ell h) + B_\ell(x) X^m(\ell h + h)] + R_\ell, \quad (17)
$$

$$
\left( h = \frac{1}{N} \right).
$$

The two functions $A_\ell(x)$ and $B_\ell(x)$ are

$$
A_\ell(x) = \frac{1}{\Delta}[(\ell h + h)^m I_\ell(x) - J_\ell(x)], B_\ell(x) = \frac{1}{\Delta}[J_\ell(x) - (\ell h)^m I_\ell(x)], \quad (18)
$$

where

$$
\Delta = h^m \sum_{\zeta=1}^{m} \frac{\Gamma(m+1)}{\Gamma(\zeta+1)\Gamma(m-\zeta+1)} \ell^{(m-\zeta)}, (-N \leq \ell \leq N)
$$

and

$$
I_\ell(x) = \int_{\ell h}^{(\ell+1)h} k(|y - x|) dy, J_\ell(x) = \int_{\ell h}^{(\ell+1)h} k(|y - x|) y^m dy. \quad (19)
$$

The integral term of Equation (17) after using Equation (18) becomes:

$$
\int_{-1}^{1} k(|x - y|) X^m(y) dy = \sum_{\ell=-N}^{N} D_\ell(x) X^m(\ell h)
$$

$$
D_\ell(x) = \begin{cases} A_{-N}(x) & , \ell = -N \\ A_n(x) + B_{n-1}(x) & , -N < \ell < N \\ B_{N-1}(x) & , \ell = N. \end{cases} \quad (20)
$$

Equations (18) and (20) are used in $x = jh, (-N \leq j \leq N)$, to obtain

$$
X(jh) = X_j, A_\ell(jh) = A_{\ell,j}, B_\ell(jh) = B_{\ell,j}, D_\ell(jh) = D_{\ell,j}, H(jh, t) = H_j(t). \quad (21)
$$

The nonlinear integral in Equation (15) shows the NAS of a $(2N + 1)$ system

$$
\mu X_j - \rho(t) \sum_{\ell=-N}^{N} D_{\ell,j} X_\ell^m = H_j(t), \quad -N \leq \ell, j \leq N. \quad (22)
$$

The matrices $D_{n,\ell}$ show the Toeplitz matrix as:

$$D_{\ell,j} = V_{n-\ell} - U_{n,\ell'} \quad V_{\ell-j} = A_{\ell,j} + B_{\ell-1,j}, \quad (\ell \le N, -N \le j),$$

$$U_{l,j} = \begin{cases} B_{-N-1,j}\ell & = -N \\ 0 & -N < \ell < N \\ A_{N,j}\ell & = N. \end{cases} \tag{23}$$

Equation (23) shows the matrix of two types with a $(2N + 1)$ order, including $V_{\ell-j}$ (Toeplitz matrix) and $U_{\ell_j}$, which shows the zero elements, excluding the first and last columns or rows. The error $R_{n,\ell}$ can be obtained based on the following formula:

$$R_\ell = \max_{0 \le j \le N} \left| \int_{\ell h}^{\ell h + h} y^{2m} k(|x - y|) dy - \left( G_\ell(x)(\ell h)^{2m} + H_\ell(x)(\ell h + h)^{2m} \right) \right|$$

$$= O\left( h^{3m} \right), \quad (x = jh). \tag{24}$$

### 7. The Nonlinear Algebraic Toeplitz Matrix System

This section provides the existence based on NAS (22) using the Banach space $\ell^\infty \times C[0, T]$. The operator form is written as:

$$\bar{T} X_j = T X_j + \frac{H_j}{\mu}, \tag{25}$$

where

$$T X_j = \frac{\rho(t)}{\mu} \sum_{\ell=-N}^{N} D_{\ell,j} X_\ell^m \quad ; \quad (-M \le m \le M, 0 \le t \le T < 1). \tag{26}$$

The following lemma is

**Lemma 3.** *If the position kernel satisfies the conditions below:*

$$(i) \left( \int_{\ell h}^{\ell h + h} \int_{jh}^{jh + h} \left\{ k^2(|x - y|) \right\} dx dy \right)^{\frac{1}{2}} \le C,$$

$$(ii) \lim_{x' \to x} \left\| k(x', y) - k(x, y) \right\|_{L_2} = 0 \quad , x, x' \in (-1, 1). \tag{27}$$

*Then,*

$$(a) \sup_N \sum_{j=-N}^{N} \left| D_{\ell,j} \right| exists,$$

$$(b) \lim_{\ell' \to l} \sup_N \sum_{j=-N}^{N} \left| D_{\ell',j} - D_{\ell,j} \right| = 0. \tag{28}$$

**Proof.** From Equations (18) and (19), we have

$$|A_\ell(x)| \le \frac{1}{|\Delta|} \left[ \left| (\ell h + h)^m \right| \left| \int_{\ell h}^{(\ell+1)h} k(|y - x|) dy \right| + \left| \int_{\ell h}^{(\ell+1)h} k(|y - x|) y^m dy \right| \right]$$

Applying the Cauchy–Schwarz inequality and taking the sum from $\ell = -N$ to $\ell = N$, the above inequality yields

$$\sum_{\ell=-N}^{N} |A_\ell(x)| \le \frac{1}{|\Delta|} \|k(|x - y|)\| \left[ \sum_{\ell=-N}^{N} \left| (\ell h + h)^m \right| + \|y^m\| \right]$$

Based on Equation (27), the function continuity $y^m$ in input $(-1, 1)$, a small constant $E_1$ exists, i.e., $\sum_{\ell=-N}^{N} |A_\ell(x)| \leq E_1$, $\forall N$. As each value of $\sum_{\ell=-N}^{N} |A_\ell(x)|$ is bounded, take $x = jh$ as:

$$\sup_N \sum_{\ell=-N}^{N} |A_\ell(jh)| \leq E_1. \tag{29}$$

Similarly, by taking a small value of the constant $E_2$ for Equations (18) and (19), we have

$$\sup_N \sum_{n=-N}^{N} |B_n(mh)| \leq E_2. \tag{30}$$

Using (29) and (30), we have

$$\sup_N \sum_{\ell=-N}^{N} \left|D_{\ell,j}\right| \leq \sup_N \sum_{\ell=-N}^{N} |A_\ell(jh)| + \sup_N \sum_{\ell=-N}^{N} |B_\ell(jh)| \leq E.$$

Hence, $\sup_N \sum_{\ell=-N}^{N} \left|D_{\ell,j}\right|$ exists.

To prove the second equation of (28) for $x, x' \in (-1, 1)$, the Cauchy–Schwarz inequality is applied by taking the sum from $\ell = -N$ to

$$\sup_N \sum_{\ell=-N}^{N} |A_\ell(x') - A_\ell(x)| \leq \frac{1}{|\Delta|} \|k(x', y) - k(x, y)\|_{L_2} \left\{ \sup_N \sum_{\ell=-N}^{\ell} [|(\ell h + h)^m| + \|y^m\|] \right\} \tag{31}$$

Using $x = jh$, $x' = j'h$ and Equation (27), we obtain that as $x' \to x$,

$$\lim_{j'j} \sup_N \sum_{\ell=-N}^{N} \left|A_\ell(j'h) - A_\ell(jh)\right| = 0. \tag{32}$$

Similarly, from (18) and (19), it is proved that

$$\lim_{j' \to m} \sup_N \sum_{t=-N}^{N} \left|B_\ell(j'h) - B_\ell(jh)\right| = 0. \tag{33}$$

Finally, we obtain

$$\lim_{m \to m} \sup_N \sum_{n=-N}^{N} |D_{m'n} - D_{mn}| = 0.$$

□

Now, the principal theorem is proven based on the nonlinear algebraic systems.

**Theorem 3.** *The NAS (22) using the Banach $\ell^\infty \times C[0, T]$ space shows the unique form of the solution, as follows:*

$$\sup_j \left|H_j(t)\right| \leq \bar{H} < \infty, \tag{34}$$

$$\sup_N \sum_{\ell=-N}^{N} \left|D_{\ell,J}\right| \leq \bar{E}, \tag{35}$$

*where $\bar{H}$ and $\bar{E}$ are constants.*

The functions $X^m(jh)$, where $m = 1, 2, \ldots, M$ for the constants $\bar{Q} > \bar{Q}_1, \bar{Q} > \bar{P}_1$ satisfy the following:

$$\sup_j |X^m(jh)| \leq \overline{Q_1} \|X\|_{\ell^\infty} \tag{36}$$

$$\sup_j |X^m(jh) - Z^m(jh)| \leq \bar{P}_1 \|X - Z\|_{\ell^\infty} \tag{37}$$

where $\|X\|_{\ell^\infty} = \sup_j |X_j|$, $X(jh) = X_j$ for each integer $j$.

The below lemmas must be proven for the above theorem.

**Lemma 4.** *If Equations (34)–(36) are tested, then the $\bar{T}$ operator is defined by using Equation (25), which maps the $\ell^\infty \times C[0, T]$ space onto itself.*

**Proof.** Suppose $U$ shows all the functions set as $X = \{X_j\}$ in $\ell^\infty \times C[0, T]$, i.e., $\|\Phi\|_{\ell^\infty \times C[0,T]} \leq \bar{\beta}, \bar{\beta}$ is constant and the $\bar{T}\Phi$ operator is based on the Banach $\ell^\infty \times C[0, T]$ space:

$$\|\bar{T}X\|_{\ell^\infty \times C[0,T]} = \sup_j |\bar{T}X_j|, \text{ for all } j$$

Using conditions (34) and (35), we have

$$|\bar{T}X_j| \leq \left|\frac{\rho(t)}{\mu}\right| Q \|X\|_{\ell^\infty} \sup_j \sum_{\ell=-N}^{N} |D_{\ell,j}| + \sup_j \left|\frac{\bar{H}}{\mu}\right|.$$

For each integer $j$, the above inequality is shown as:

$$\sup_j |\bar{T}X_j| \leq \sigma_1 \|X\|_{\ell^\infty} + \frac{\bar{H}}{\mu}, \left(\sigma_1 = \left|\frac{\rho}{\mu}\right| \bar{Q}\bar{E}\right) \tag{38}$$

The above inequality (38) represents that the $\bar{T}$ operator maps the $U$ set, where

$$\bar{\beta} = \frac{\bar{H}}{(|\mu| - |\rho|\bar{Q}\bar{E})}.$$

Hence, $\sigma_1 < 1$ have been taken, whereas $T$ and $\bar{T}$ operators are bounded. $\square$

**Lemma 5.** *Under the conditions (34), (35) and (37), $\bar{T}$ shows a contraction operator in the $\ell^\infty \times C[0, T]$ space*

**Proof.** For $X$ and $Z$ functions in $\ell^\infty \times C[0, T]$, Formulas (25) and (26) become:

$$|\bar{T}X_j - \bar{T}Z_j| \leq \left|\frac{\rho}{\mu}\right| \sum_{n=-N}^{N} |D_{\ell,j}| \sup_j |X_j - Z_j|.$$

Using conditions (35) and (37),

$$\|\bar{T}X - \bar{T}Z\|_{\ell^\infty \times C[0,T]} \leq \sigma_1 \|X - Z\|_{\ell^\infty \times C[0,T]}, \left(\sigma_1 = \left|\frac{\rho}{\mu}\right| \bar{Q}\bar{E}\right). \tag{39}$$

The above form indicates that the $\bar{T}$ operator is continuous in $\ell^\infty \times C[0, T]$ space. $\bar{T}$ shows the contraction operator based on $\sigma_1 < 1$. Hence, $\bar{T}$ presents a unique fixed point with specific solutions of the NAS using $\ell^\infty \times C[0, T]$ space. $\square$

### 8. The Error of the Toeplitz Matrix Method

In any practical use of the TMM, some estimation of the error size is involved. Hence, these two definitions were used to calculate the error of the TMM.

**Definition 2.** *A local error $R_j$ is used as:*

$$X(x) - X_{\xi}(x) = \sum_{\ell=-N}^{N} D_{\ell,j}\left[X_j^m - X_{j,\xi}^m\right] + R_j, \, (x = jh), \tag{40}$$

*where $X_{\xi}(x)$ shows the approximate results of (2).*

In the other formula, Equation (40) is used as:

$$R_j = \left| \int_{-1}^{1} k(|y - x|)X^m(y)dy - \sum_{\ell=-N}^{N} D_{\ell,j}X_j^m \right|.$$

**Definition 3.** *The TMM shows a convergence of r order in input $(-1,1)$; conversely, by taking the large values of $N$, $\bar{D} > 0$ exists based on $N$ independently, that is,*

$$\|X(x) - X_N(x)\| \leq \bar{D}N^{-r}. \tag{41}$$

Now, we present the theorem, which is provided using NAS, based on Equation (22), which has a unique solution.

**Theorem 4.** *The error $R_j$ is considered negligible as $j \to \infty$*

$$\lim_{j \to \infty} R_j = 0. \tag{42}$$

**Proof.** Equation (40) is used as:

$$\left|R_{\xi}\right| \leq \left|X_j - (X_j)_{\xi}\right| + \sum_{\ell=-N}^{N}\left|D_{\ell,j}\right| \sup_j \left|X_j^m - \left(X_j^m\right)_{\xi}\right|.$$

Using conditions (35) and (37), along with each integer $\xi$, we obtain

$$\left\|R_{\xi}\right\|_{\ell^{\infty} \times C[0,T]} \leq (1 + EQ)\left\|X - X_{\xi}\right\|_{\ell^{\infty} \times C[0,T]}. \tag{43}$$

Since $\left\|X - X_{\xi}\right\|_{\ell^{\infty} \times C[0,T]} \to 0$ as $\xi \to \infty$, $\forall t \in [0,T]$, then $R_j \to 0$. □

### 9. Applications

For numerical results and tables, we used Maple 2022.1 software, Version 15, March 2022, Windows, 10, 8 G RAM, 64-bit. In this section, the NMIE of (1) in the following special form was considered

$$\frac{\partial^{\alpha}\Phi(x,t)}{\partial t^{\alpha}} + 0.5\Phi(x,t) = g(x,t) + 0.33\int_{-1}^{1} k(|y - x|)\Phi^m(y,t)dy, \left(\Phi(x,0) = x^2 \tag{44}\right)$$

The true result of Equation (44) is $\Phi(x,t) = \left(0.5t^{0.5} + 0.25t^{1.5} + x^2\right))$.

**Example 1** (For logarithmic kernel).

$$k(|x - y|) = \ln(|x - y|). \tag{45}$$

After using (17)–(19) including the famous integral reference as Gradstein et al. [34]:

$$\int x^m \ln(a + bx)dx = \frac{1}{m+1}\left[x^{m+1} - \frac{(-a)^{m+1}}{b^{m+1}}\right]\ln(a + bx) + \frac{1}{m+1}\sum_{k=1}^{m+1}\frac{(-1)^k x^{m-k+2}a^{k-1}}{(m-k+2)b^{k-1}}, \quad (46)$$

we have

$$\begin{aligned}
A_J(\ell h) =& \frac{h}{[J^m - (J+1)^m]}\left[\frac{1}{(m+1)}\left[\left[(J+1)^{m+1} - \ell^{m+1}\right]\ln|(\ell - J - 1)h|\right.\right. \\
&\left.- \left[J^{m+1} - \ell^{m+1}\right]\ln|(\ell - J)h|\right] + (J+1)^m[(\ell - J - 1)\ln|(\ell - J - 1)h| - (\ell - J)\ln|(\ell - J)h| + 1] \\
&\left.- \frac{1}{(m+1)}\sum_{k=1}^{m+1}\frac{\left[(J+1)^{m-k+2} - J^{m-k+2}\right]\ell^{k-1}}{(m-k+2)}\right]
\end{aligned} \quad (47)$$

and

$$\begin{aligned}
B_J(\ell h) =& \frac{h}{[(J+1)^m - J^m]}\left\{\frac{1}{(m+1)}\left[\left[(J+1)^{m+1} - \ell^{m+1}\right]\ln|(\ell - J - 1)h| - \left[J^{m+1} - \ell^{m+1}\right]\ln|(\ell - J)h|\right]\right\} \\
&+ \frac{h}{[(J+1)^m - J^m]}\{J^m[(\ell - J - 1)\ln|(\ell - J - 1)h| - (\ell - J)\ln|(\ell - J)h| + 1] \\
&\left.- \frac{1}{(m+1)}\sum_{k=1}^{m+1}\frac{\left[(J+1)^{m-k+2} - J^{m-k+2}\right]\ell^{k-1}}{(m-k+2)}\right\}.
\end{aligned} \quad (48)$$

Using (47) and (48), the coefficients $D_{\ell J}$ of the nonlinear algebraic system (22) are presented as:

$$\begin{aligned}
D_{\ell J} =& \frac{h}{[J^m - (1+J)^m]}\left\{(1+m)^{-1}\left[\left[(1+J)^{m+1} - \ell^{m+1}\right]\ln|(\ell - 1 - J)h| - \left[J^{m+1} - \ell^{m+1}\right]\ln|(\ell - J)h|\right]\right. \\
&+ (J+1)^m[(\ell - J - 1)\ln|(\ell - J - 1)h| - (\ell - J)\ln|(\ell - J - 1)h| + 1] \\
&\left.- \frac{1}{(m+1)}\sum_{k=1}^{m+1}\frac{\left[(J+1)^{m-k+2} - J^{m-k+2}\right]\ell^{k-1}}{(m-k+2)}\right. \\
&+ \frac{h}{[J^m - (J-1)^m]}\left\{\frac{1}{(1+m)}\left[\left[J^{m+1} - \ell^{m+1}\right]\ln|(\ell - J)h| - \left[(J-1)^{m+1} - \ell^{m+1}\right]\ln|(\ell - J + 1)h|\right]\right\}.
\end{aligned} \quad (49)$$

In addition,

$$R = \left|\int_a^{a+h}\ln|x - y|\varphi^i(y)dy - A_n(x)\varphi^i(a) - B_n(x)\varphi^i(a + h)\right|. \quad (50)$$

The formula (50) takes the form:

$$\begin{aligned}
R =& Ch^{2m+1}, \\
C =& \left|\left(\frac{\ell^{2m+1}}{2m+1} - \frac{\ell^{m+1}}{m+1}\right)\ln|\ell h| - \left(\frac{\ell^{2m+1} - 1}{2m+1} - \frac{\ell^{m+1} - 1}{m+1}\right)\ln|h(\ell - 1)|\right. \\
&\left.- \sum_{k=1}^{2m+1}\frac{\ell^{k-1}}{(2m-k+2)(2m+1)} + \sum_{k=1}^{m+1}\frac{\ell^{k-1}}{(m-k+2)(m+1)}\right|.
\end{aligned} \quad (51)$$

The linear case can be obtained from Equations (47)–(51) by letting $m = 1$.

**Example 2** (For Carleman kernel)**.**

$$k(|x - y|) = |x - y|^{-\beta}(0 < \beta < 1). \quad (52)$$

*The significance based on the Carleman kernel was shown in the Arutiunian [35] work that presented the plane contact problem using nonlinear plasticity theory as its first calculation. It was reduced through the first-order Fredholm integral model with a Carleman kernel.*

After using (17) to (19), we obtain

$$
A_J(\ell h) = \frac{h^{1-\beta}}{J^m - (1+J)^m} \left\{ \sum_{k=0}^{m} \frac{m! \left[ J^{m-k} |\ell - J|^{k+1-\beta} - (J+1)^{m-k} |\ell - J - 1|^{k+1-\beta} \right]}{(m-k)!(1-\beta)(2-\beta)\dots(k+1-\beta)} \right.
$$
$$
\left. + \frac{(1+J)^m}{(-\beta+1)} \left[ |\ell - J - 1|^{1-m} - |\ell - J|^{1-m} \right] \right\}
$$
(53)

and

$$
B_J(\ell h) = \frac{h^{1-\beta}}{[-J^m + (1+J)^m]} \left\{ \sum_{k=0}^{m} \frac{m! \left[ J^{m-k} |\ell - J|^{k+1-\beta} - (J+1)^{m-k} |\ell - J - 1|^{k+1-\beta} \right]}{(m-k)!(1-\beta)(2-\beta)\dots(k+1-\beta)} \right.
$$
$$
\left. + \frac{J^m}{(1-\beta)} \left[ |\ell - J - 1|^{1-\beta} - |\ell - J|^{1-\beta} \right] \right\}
$$
(54)

Therefore, the Toeplitz matrix $D_{\ell,J}$ becomes:

$$
D_{\ell,J} = h^{1-\beta} \left\{ \frac{1}{[J^m - (J+1)^m]} \left[ \sum_{k=0}^{m} \frac{m! [J^{m-k} |\ell - J|^{k+1-\beta} - (J+1)^{m-k} |\ell - J - 1|^{k+1-\beta}]}{(m-k)!(1-\ell)(2-\beta)\dots(k+1-\beta)} + \frac{(J+1)^m}{(1-\beta)} \left[ |\ell - J - 1|^{1-\beta} - |\ell - J|^{1-\beta} \right] \right] \right.
$$
$$
\left. + \frac{1}{[[J^m - (J-1)^m]]} \left[ \sum_{k=0}^{m} \frac{m! [(J-1)^{i-k} |\ell - J + 1|^{k+1-\beta} - J^{m-k} |\ell - J|^{k+1-\beta}]}{(m-k)!(1-\beta)(2-\beta)\dots(k+1-\beta)} + \frac{(J-1)^i}{(1-\beta)} \left[ |\ell - J|^{1-\beta} - |\ell - J + 1|^{1-\beta} \right] \right] \right\}
$$
(55)

The error $R$ is shown as:

$$
|R| \le Ch^{2m+1-\beta},
$$

$$
C = \left| \sum_{k=0}^{m} \frac{m! \ell^{k+1-\beta} \left| 1 - \frac{1}{\ell} \right|^{k+1-\beta}}{(m-k)!(1-\beta)(2-\beta)\dots(k+1-\beta)} - \sum_{k=0}^{2m} \frac{(2m)! \ell^{k+1-\beta} \left| 1 - \frac{1}{\ell} \right|^{k+1-\beta}}{(2m-k)!(1-\beta)(2-\beta)\dots(k+1-\beta)} \right|.
$$
(56)

**Example 3** (Suppose the Hilbert kernel).

$$
k(|x-y|) = \cot\left( \left| \frac{x-y}{2} \right| \right), \varphi(\pm \pi, t) = 0.
$$
(57)

*The exact output of Equation (57) is $\varphi(x,t) = (0.5t^{0.5} + 0.25t^{1.5}) \sin x$.*

*The integral equation based on the Hilbert kernel, together with the crack problem used in elasticity theory are discussed in [34].*

$$
\int_{nh}^{(n+1)h} x^m \cot x \, dx = \sum_{s=0}^{\infty} \frac{(-1)^{s^{2s}} 2s B_{2s}}{(m+2s)(2s)!} x^{m+2s} \quad (m \ge 1, |x| < \pi).
$$
(58)

where Bernoulli numbers $B_{2s}$ is used as:

$$
D_{J,\ell} = 2 \left\{ (\ell - J + 1) \ln \left| \sin \frac{h(\ell - J + 1)}{2} \right| - 2(\ell - J) \ln \left| \sin \frac{h(\ell - J)}{2} \right| + (\ell - J - 1) \ln \left| \sin \frac{h(\ell - J - 1)}{2} \right| \right\}
$$
$$
- \sum_{s=0}^{\infty} \frac{(-1)^s 2^{2s} B_{2s}}{(m+2s)(2s)!} \times \left\{ (\ell - J + 1)^{1+2s} - 2(\ell - J)^{1+2s} + (\ell - J - 1)^{1+2s} \right\}, m = 1, 2, \dots, M.
$$

From the above, the results of using the logarithmic function and the effect of time in the first example, as well as the arithmetic error, are described numerically in Table 1 and

Figure 1a,b in the nonlinear case ($m = 2$) and linear case ($m = 1$). In the second example, the results of using the Carleman function over time periods for the nonlinear and linear cases, as well as the resulting arithmetic error, were derived in Table 2 and Figure 2a,b, respectively. While Table 3 and Figure 3a,b represent the numerical results and arithmetic errors corresponding to the nonlinear and linear cases, respectively, for different values of the coefficient of the Carleman function. In the third example of the Hilbert kernel, the numerical error of the nonlinear case and the linear case has been calculated at $t = 0.1$ in Figure 4a,b, respectively. While for $t = 0.4$ the corresponding results of errors are described in Figure 5a,b.

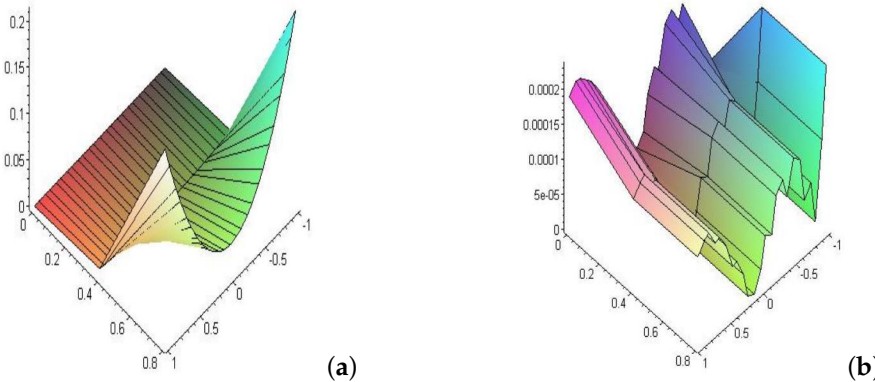

**Figure 1.** (**a**) Shows the corresponding error of NMIE with a logarithmic kernel at different times, whereas (**b**) illustrates the corresponding error for the linear case at the same times.

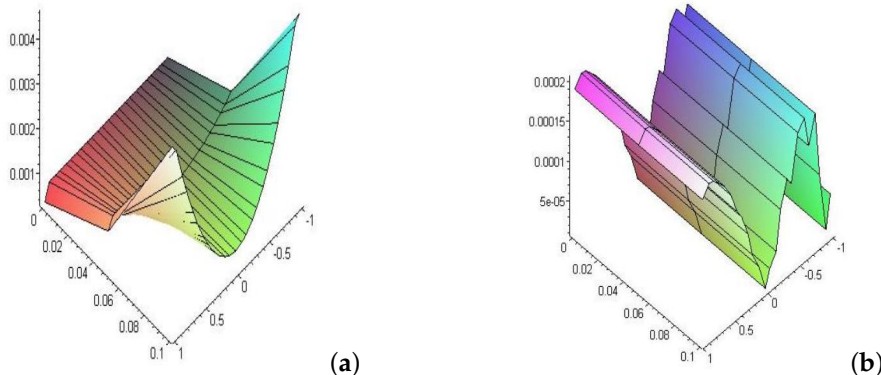

**Figure 2.** (**a**) Shows the error for the nonlinear case at $\beta = 0.01$ and (**b**) shows the corresponding error for the linear case at $\beta = 0.01$.

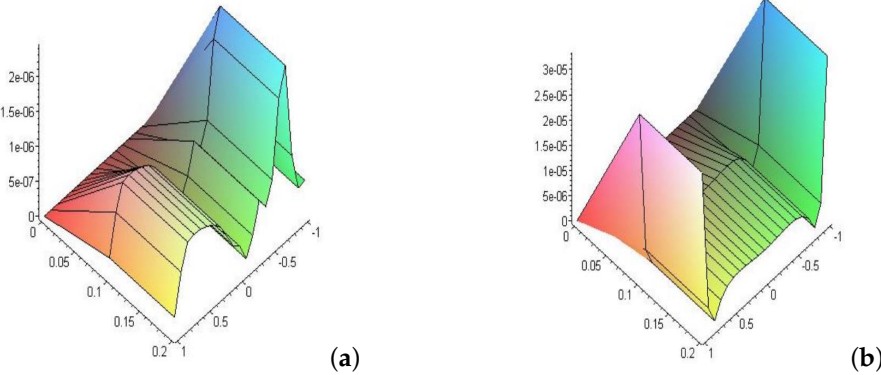

**Figure 3.** (**a**,**b**) Shows the error of Equation (44) respectively for the nonlinear and linear cases of Carleman coefficients at t = 0.2 and N = 20.

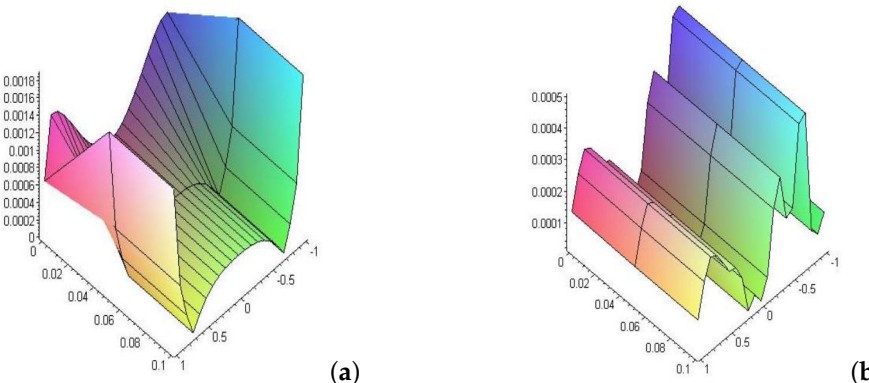

**Figure 4.** (**a**,**b**) Shows the error of Equation (44) using the Hilbert kernel for the nonlinear and the linear cases at time $t = 0.1$, N = 20.

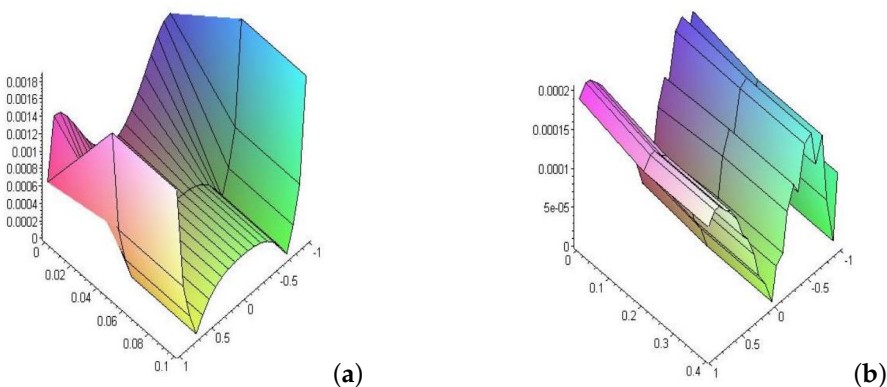

**Figure 5.** (**a**,**b**) Shows the error of Equation (44) with Hilbert kernel for the nonlinear and the linear cases at time t = 0.4, N = 20.

**Table 1.** Solutions of the linear/nonlinear solutions of MIE (44) with a logarithmic kernel, along with the corresponding errors using TMM.

| Time | Exact Solution | Nonlinear Case m = 2, N = 20 | | Linear Case m = 1, N = 20 | |
|---|---|---|---|---|---|
| $t$ | $\varphi_{\text{Exact}}$ | $\varphi_{\text{Non.}}$ | $E_{\text{Non.}}$ | $\varphi_{\text{Lin.}}$ | $E_{\text{Lin.}}$ |
| $t = 0.01$ | 1.0001000 | 1.00010270 | $0.269 \times 10^{-7}$ | 1.00010561 | $0.567 \times 10^{-7}$ |
| | 0.9345000 | 0.93450708 | $0.362 \times 10^{-7}$ | 0.93450758 | $0.384 \times 10^{-7}$ |
| | 0.4012111 | 0.40122142 | $0.496 \times 10^{-7}$ | 0.40122136 | $0.187 \times 10^{-7}$ |
| | 0.0901000 | 0.09010238 | $0.976 \times 10^{-7}$ | 0.09010146 | $0.705 \times 10^{-7}$ |
| $t = 0.1$ | 1.0100000 | 1.01003928 | $0.286 \times 10^{-6}$ | 1.01006525 | $0.525 \times 10^{-6}$ |
| | 0.9444440 | 0.94444790 | $0.324 \times 10^{-6}$ | 0.94444837 | $0.392 \times 10^{-6}$ |
| | 0.6500000 | 0.65000474 | $0.297 \times 10^{-6}$ | 0.65000101 | $0.898 \times 10^{-6}$ |
| | 0.5977778 | 0.59779381 | $0.160 \times 10^{-6}$ | 0.59778257 | $0.172 \times 10^{-6}$ |
| $t = 0.4$ | 1.1600000 | 1.60004729 | $0.292 \times 10^{-5}$ | 1.16000215 | $0.215 \times 10^{-5}$ |
| | 1.09444440 | 1.09444173 | $0.434 \times 10^{-5}$ | 1.09444627 | $0.182 \times 10^{-5}$ |
| | 0.7477778 | 0.74772763 | $0.288 \times 10^{-5}$ | 0.74779495 | $0.982 \times 10^{-6}$ |
| | 0.6977778 | 0.69777818 | $0.341 \times 10^{-6}$ | 0.69777785 | $0.231 \times 10^{-5}$ |
| $t = 0.8$ | 1.640000 | 1.64007850 | $0.211 \times 10^{-5}$ | 1.64001375 | $0.137 \times 10^{-5}$ |
| | 1.177778 | 1.17784366 | $0.588 \times 10^{-5}$ | 1.17781185 | $0.340 \times 10^{-5}$ |
| | 0.711111 | 0.71115465 | $0.559 \times 10^{-5}$ | 0.71118519 | $0.740 \times 10^{-5}$ |

**Table 2.** Describes the nonlinear and linear cases for MIE (44), with the Carleman kernel, time and $\beta = 0.01$.

| Time | Exact Solution | Nonlinear Case m = 2, N = 20 | | Linear Case m = 1, N = 20 | |
|---|---|---|---|---|---|
| $t$ | $\varphi_{\text{Exact}}$ | $\varphi_{\text{Non.}}$ | $E_{\text{Non.}}$ | $\varphi_{\text{Lin.}}$ | $E_{\text{Lin.}}$ |
| $t = 0.01$ | 1.0001000 | 1.0001000 | $0.545 \times 10^{-8}$ | 1.0001000 | $0.567 \times 10^{-8}$ |
| | 0.6945000 | 0.6945000 | $0841 \times 10^{-8}$ | 0.6945000 | $0.956 \times 10^{-8}$ |
| | 0.0178777 | 0.0178777 | $0.708 \times 10^{-8}$ | 0.0178777 | $0344 \times 10^{-8}$ |
| | 0.0278777 | 0.0278777 | $0.710 \times 10^{-8}$ | 0.0278777 | $0.547 \times 10^{-8}$ |
| $t = 0.1$ | 1.0100000 | 1.0100000 | $0.456 \times 10^{-7}$ | 1.0100000 | $0.648 \times 10^{-7}$ |
| | 0.9444444 | 0.9444448 | $0.465 \times 10^{-7}$ | 0.9444447 | $0.375 \times 10^{-7}$ |
| | 0.6500000 | 0.6500065 | $0.342 \times 10^{-7}$ | 0.6500004 | $0.173 \times 10^{-7}$ |
| | 0.2600000 | 0.2600008 | $0.179 \times 10^{-7}$ | 0.2600008 | $0.160 \times 10^{-7}$ |
| $t = 0.4$ | 1.1600000 | 1.1600875 | $0.700 \times 10^{-5}$ | 1.1600075 | $0.750 \times 10^{-5}$ |
| | 0.8000000 | 0.8000373 | $0.753 \times 10^{-5}$ | 0.8000290 | $0.251 \times 10^{-5}$ |
| | 0.6044444 | 0.604439 | $0.194 \times 10^{-5}$ | 0.6044499 | $0.244 \times 10^{-5}$ |
| | 0.2711111 | 0.2711951 | $0.832 \times 10^{-5}$ | 0.2711990 | $0.612 \times 10^{-5}$ |
| $t = 0.8$ | 1.6400000 | 1.6400332 | $0.331 \times 10^{-4}$ | 1.6400542 | $0.542 \times 10^{-4}$ |
| | 1.0844444 | 1.0849735 | $0.549 \times 10^{-4}$ | 1.0849810 | $0.416 \times 10^{-4}$ |
| | 0.7300000 | 0.7304171 | $0.416 \times 10^{-4}$ | 0.73082523 | $0.311 \times 10^{-4}$ |
| | 0.6444444 | 0.6443994 | $0.053 \times 10^{-4}$ | 0.64440364 | $0.170 \times 10^{-5}$ |

**Table 3.** Shows the nonlinear and linear cases for Equation (44) for different values of Carleman coefficients at time $t = 0.2$, N = 20.

| Carleman Coefficients $\beta$ | Exact Solution at $t = 0.2$ | Nonlinear Case at m = 2 | | Linear Case at m = 1 | |
|---|---|---|---|---|---|
| | $\varphi_{\text{Exact}}$ | $\varphi_{\text{Non.}}$ | $E_{\text{Non.}}$ | $\varphi_{\text{Lin.}}$ | $E_{\text{Lin.}}$ |
| $\beta = 0.22$ | 0.0400000 | 0.0400099 | $0.519 \times 10^{-6}$ | 0.04003999 | $0.199 \times 10^{-6}$ |
| | 0.0096040 | 0.00960747 | $0.467 \times 10^{-6}$ | 0.00965943 | $0.624 \times 10^{-6}$ |
| | 0.4000000 | 0.40000256 | $0.166 \times 10^{-6}$ | 0.40000373 | $0.373 \times 10^{-6}$ |
| | 0.0400000 | 0.04000081 | $0.158 \times 10^{-6}$ | 0.0400006 | $0.167 \times 10^{-6}$ |
| $\beta = 0.32$ | 0.0400000 | 0.04000220 | $0.822 \times 10^{-5}$ | 0.04003999 | $0.890 \times 10^{-5}$ |
| | 0.0163840 | 0.01632257 | $0.798 \times 10^{-5}$ | 0.01631233 | $0.466 \times 10^{-5}$ |
| | 0.0010240 | 0.00102484 | $0.364 \times 10^{-5}$ | 0.00102149 | $0.260 \times 10^{-5}$ |
| | 0.0400000 | 0.04000523 | $0.220 \times 10^{-5}$ | 0.04000061 | $0.111 \times 10^{-5}$ |
| $\beta = 0.7$ | 0.0400000 | 0.04030011 | $0.434 \times 10^{-4}$ | 0.04000210 | $0.210 \times 10^{-5}$ |
| | 0.0010240 | 0.00100645 | $0.754 \times 10^{-5}$ | 0.00100266 | $0.262 \times 10^{-5}$ |
| | 0.0163840 | 0.01638701 | $0.197 \times 10^{-5}$ | 0.01638794 | $0.457 \times 10^{-5}$ |
| | 0.0400000 | 0.04003430 | $0.430 \times 10^{-5}$ | 0.04999902 | $0.973 \times 10^{-6}$ |
| $\beta = 0.8$ | 0.0400000 | 0.04000037 | $0.375 \times 10^{-5}$ | 0.04000021 | $0.219 \times 10^{-6}$ |
| | 0.0096040 | 0.00960777 | $0.237 \times 10^{-5}$ | 0.00960137 | $0.974 \times 10^{-6}$ |
| | 0.0000640 | 0.00006437 | $0.243 \times 10^{-5}$ | 0.00006407 | $0.292 \times 10^{-5}$ |
| | 0.0163840 | 0.016385043 | $0.112 \times 10^{-5}$ | 0.01638486 | $0.214 \times 10^{-5}$ |

## 10. Conclusions

The following conclusions were drawn:

1.  In this paper, the existence of a unique solution is proven using the Banach fixed point theorem. In addition, the reader could use the successive approximate method (Picard method) to arrive at the same conclusion. In the homogeneous case of Equation (1), the successive approximate method fails to prove the existence of a unique solution. For this, we can only use the Banach fixed point theorem.

2. If the two conditions of (i) and (ii) are not satisfied, this means that we have at least one solution. In this case, we would use one of the following theorems: Brouwer fixed point theorem or Schauder fixed point theorem.

3. Using TMM, we have an NAS where the coefficient of the nonlinear term is a function of time. Hence, the existence of a unique solution for the NAS is discussed in the space $\ell^\infty \times C[0, T], T < 1$.

4. The fractional nonlinear mixed integro-differential Equation (1), under certain relations of $\mu$ and $v$, represents the nonlinear integral equation of the fractional phase-lag term

$$v\Phi(x, t + \delta t) = g(x, t) + \lambda \int_\Omega k(|x - y|)\Phi^m(y, t)dy, \quad \left(\mu = v\frac{(\delta t)^\alpha}{\Gamma(\alpha)}\right). \quad (59)$$

The delaying or advancing of time reveals the natural phenomena, especially in the presence of thermoelectricity and magnetic media. Some of applications of fractional integro-differential equations are found in physics, chemistry, economics, and biology [12,29]. Equation (59) explains the physical meaning of the fractional equation of time as the first fractional approximation of the time lag equation, and this lag may be before or after real time.

5. The variable separation technique used in Equation (11) enabled the researchers to find the necessary time relationship between the nucleus and time, as presented in Equation (15).

6. The significance of the logarithmic kernel was approved from its derivatives $f$ with these cases:

    (a) $\frac{\partial}{\partial x}k(|y - x|) = \left(\frac{1}{|y - x|}\right)$ Cauchy kernel.

    (b) $\frac{\partial^2}{\partial x^2}k(|y - x|) = \left(\frac{1}{|y - x|^2}\right)$ Strong singular kernel

    (c) The Carleman function was also established as:

$$\ln|y - x| = \underbrace{\left[(\ln|y - x|)|y - x|^v\right]}_{U(y,x)} |y - x|^{-v}$$

where $U(y, x)$ is a continuous function.

7. When the kernel of the equation was in the logarithmic function form $k(|x - y|) = \ln(|x - y|)$, the relative error increased with increasing time. It was also noted that the error in the non-linear case was slightly larger than in the linear case.

8. In Example (2), when the kernel took the Carleman function $k(|x - y|) = |x - y|^{-\beta}(0 < \beta < 1)$, we noticed that the behavior of the error when increasing time was the same as that of the logarithmic function. However, by comparison, we found that at small times, the error in the logarithmic function was higher than in the Carleman function. With increasing time, we find that the relative error in the Carleman function is higher than its counterpart in the logarithmic function.

9. In Example (3), the error behavior of the Hilbert kernel $k(|x - y|) = \cot\left(\left|\frac{x-y}{2}\right|\right)$ was the same as that of the logarithmic form and Carleman function.

## 11. Future Work

Future work will attempt to solve Equation (1) when the coefficients of the equation are variable. This will lead to solutions for many applications in the sciences related to nonlinear elasticity.

**Author Contributions:** Supervision, M.A.A.; Project administration, S.E.A. All authors have read and agreed to the published version of the manuscript.

**Funding:** This research was funded by the project number: 22UQU4282396DSR01, through the Deanship for Research & Innovation, Ministry of Education in Saudi Arabia.

**Institutional Review Board Statement:** Not applicable.

**Informed Consent Statement:** Not applicable.

**Data Availability Statement:** Data is contained within the article.

**Acknowledgments:** The authors would like to thank the reviewers for their suggestions that helped improve the research. The authors thank the Deanship for Research & Innovation, Ministry of Education in Saudi Arabia for funding this research work through the project number: 22UQU4282396DSR01.

**Conflicts of Interest:** The funders had no role in the design of the study; in the collection, analyses, or interpretation of data; in the writing of the manuscript; or in the decision to publish the results.

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
