# Peer review of "A Physical Phenomenon for the Fractional Nonlinear Mixed Integro-Differential Equation Using a General Discontinuous Kernel"

_fractalfract, doi:10.3390/fractalfract7020173_

Round 1
Reviewer 1 Report
I found this paper too difficult to understand because it was badly formatted and problem statement and background is not clear for readers. Authors must duly compile it using contemporary latex template and formulate problems and results with proofs.
Author Response
|
Page number
|
The part to be modified
|
New status
|
|
|
|
|
|
|
Abstract |
|
|
Page 1 line 2 |
--is presented in a general--- |
is presented and has a general |
|
Line 3 |
The existence and uniqueness of the solutions |
The conditions of existence and uniqueness solution is provided |
|
Line 4 |
The values of the Fr-NMIDE have been used to apply the properties of fractional integral, second order Volterra – Hammerstein integral equation |
After applying the properties of fractional integral, the Fr-NMIDE conformed to Volterra – Hammerstein integral equation (V- HIE) of the second kind |
|
Lines 5,6 |
The separation method is applied to the Hammerstein integral equation along with the physical coefficients. |
Then, using a technique of separating method we have HIE, where its physical coefficients are variable in time. |
|
Lines 8-11 |
Toeplitz matrix scheme is used to the nonlinear algebraic system along with the discussions of convergent. |
Toeplitz matrix method (TMM) and its scheme is used to obtain a nonlinear algebraic system with studying the convergent of the system. |
|
|
Keywords: |
|
|
|
Integro differential model |
Integro differential equation |
|
|
Introduction |
|
|
Page 1 |
As integro-differential equations (IDEs) can be used to simulate a wide range of physical issues, numerous scholars have focused a great deal of attention to present the solution of these systems. |
Because integro-differential equations (IDEs) can be used to simulate a wide range of problems in the basic sciences, many scientists have focused a great deal of attention on presenting the solution of these systems.
|
|
Page 1 line 1and 4 from below |
(i)The linear/nonlinear IDEs (ii) for solving the IDEs |
(i) The linear/nonlinear IEs / IDEs (ii) for solving the IEs / IDEs |
|
Page 2 |
generalized fractional thermoelasticity model [11], thermoelasticity mathematical with phase- lag [12-13] |
References [11,12,13] and their comments have been omitted because they are not related to the research specialization
|
|
Page 2 line 5 |
Orthogonal polynomials method is considered one |
Orthogonal polynomials method is considered as one |
|
Page 2 line 7 |
a new technique based on the separation of variables and the orthogonal polynomials method |
a new technique based on separation of variables and orthogonal polynomials method |
|
Page 2 line 15 |
Abdou and Awad [23] … to discuss the mixed integral equation using the potential kernel. |
Abdou and Awad [23] … to discuss the solution of mixed integral equation with potential kernel. |
|
Page 2 line 16 |
Abdou et al. [24] discussed the Chebyshev polynomials |
Abdou et al. [24] used Chebyshev polynomials ---- |
|
Page 2 line 18 |
Basseem and Alalyani [25] used Chebyshev polynomials to get the numerical performances of the quadratic integral model based logarithmic kernel. |
Basseem and Alalyani [25] used Chebyshev polynomials to discuss the numerical solution of the quadratic integral equation with logarithmic kernel. |
|
Page 2 line 11 from below |
Almasieh and Meleh [28] applied the hybrid function scheme to demonstrate the nonlinear form of the integral model using the continuous Fredholm kernel |
Almasieh and Meleh [28] applied the hybrid function scheme to demonstrate the nonlinear Fredholm integral model with continuous kernel |
|
Page 2 line 8 from below |
integral model based continuous kernel. |
integral model has a continuous kernel. |
|
Page 2 line 7 from below |
by finding the approximate results based on the second order Volterra integral using the discontinuous kernels. |
to find the approximate results based on the Volterra integral equations of the second kind have discontinuous kernels. |
|
Page 2 line 5 from below |
the numerical outputs of the nonlinear integral model |
the numerical outputs of a nonlinear integral model |
|
Page 2 line 4 from below |
Tarasov [32] demonstrated the electromagnetic fields using the dielectric media, which is presented by differential models with non-integer kind of time derivative. |
Reference [32] and its comments has been omitted because it is not related to the research specialization
|
|
Page 2 line 2 from below |
Abdel-Rehim [34] provided a wide review based on the theory of continuous time random walk along with the space--time fractional diffusion process. |
Reference [34] and its comments has been omitted because it is not related to the research specialization
|
|
Page 3 line 5 |
are the known and unidentified continuous functions |
are the known and unknown continuous functions, respectively |
|
Page 3 line 7 from below |
the theorem based Banach fixed point is discussed using the existence and uniqueness of |
the theorem based Banach fixed point is discussed to prove the existence and uniqueness of |
|
|
2. Solution’s existence and uniqueness
|
|
|
Page 4 line 9
|
function will be discussed using the same space |
function will be discussed in the same space |
|
Page 4 line 5 from below |
The position meets the discontinuity in |
The position kernel satisfies |
|
Page 4 line 4 from below |
where is taken as a constant. |
(C - constant) |
|
Page 4 line 2 from below |
Therefore, the kernel of position and time satisfies the,
|
Therefore, the kernel of position and time insatisfies |
|
Page 5 line 1 |
The derived function using the partial kinds of the derivatives in position and the norm are shown as ,
|
The continuous functionand its norm is |
|
Lemma (1): Page 5 |
the conditions (1) to (3), the operator maps space as:
|
Under the conditions (i) to (iii- a), the operator maps the space into itself:
|
|
Page 5 line 5 (below) |
The Eq. (3) is used to solve |
The equation (3) is used to prove |
|
Page 5 line 3 (below) |
By using (i)- (iii-a) along with the inequality of Cauchy-Schwarz presented as:
|
Using (i)- (iii-a) and the inequality of Cauchy-Schwarz, we have |
|
|
3.Convergence of the solution |
|
|
Page 7 line2 |
Eq. (8) is updated by using the Eq. (9) as |
Equation (8) is updated by using the equation (9) as |
|
Page 7, line 8 |
Th Eq. (10) shows |
Equation (10) shows |
|
|
6. Toeplitz matrix method
|
|
|
Page 10 line 11 |
the integral term of Eq. (17) using Eq. (18-19) becomes as |
The integral term of equation (17) after using equation (18) becomes |
|
|
7. The nonlinear algebraic Toeplitz matrix system
|
|
|
Page 12 line 5 |
Bases on the Eq. (27), |
Bases on the equation (27), |
|
Page 12 line 9 |
Similarly, Eqs. (18) and (19) |
Similarly, equations (18) and (19) |
|
Page 13 Lemma 4 |
which maps space |
which maps space into itself |
|
|
References
|
|
|
|
References [1,2,3,4,6 ] have been updated and written in the introduction using red color
|
|

Reviewer 2 Report
This paper focuses on the fractional nonlinear mixed integro-differential equation with discontinuous kernel. The paper format is confusion, it is necessary to use LaTeX for math papers, very difficult to follow.
comments:
1. The class of equations under study is not clear. Why authors use term "geneal" for the kernel, but study the special form?
2. What is the motivation for the research conducted?
3. The review is not sufficient. The questions of solvability of Volterra integral equations with discontinuous kernels were considered in [1]. Author must online that such equations even in linear case [1] can have non-unique solution.
4. concluding remarks are too short. what was achieved and what to do next pls clarify.
[1] Sidorov, D.N. Solvability of systems of Volterra integral equations of the first kind with piecewise continuous kernels. Russ Math. 57, 54–63 (2013). https://doi.org/10.3103/S1066369X13010064
Author Response

(The authors gave the same response as above.)

Reviewer 3 Report
Please see the attached file.

Author Response

(The authors gave the same response as above.)

Round 2
Reviewer 1 Report
Dear authors,
pls focus on the principal comments regarding the integral equations with discontinuous kernels introduced in the book: World Scientific Series on Nonlinear Science Series A: Volume 87. Integral Dynamical Models: Singularities, Signals and Control https://doi.org/10.1142/9278 | October 2014.
Author Response
Attached to your Excellency, with many thanks,
FOR you comment
Extensive editing of English language and style required
you find attachment a new copy of the article after reviewing the language from the Journal Center and also amending all recommendations

Reviewer 2 Report
In the comments regarding the paper on the Volterra integral equations of the 1st kind with with piecewise continuous kernels (which are of course the special case of discontinuous kernels!) authors confusing such kernels with continuous kernels. Authors are expected to duly read the theory of Volterra equations and review the state of the art. Regarding the theory and numerical methods authors can also refer to the seminar book by D. Sidorov Integral Dynamical Models - Singularities, Signals and Control. World Scientific Series on Nonlinear Science Series A: Volume 87, World Scientific. 2014 (https://doi.org/10.1142/9278 )
Author Response
MY dear
Warm Greetings for you and
Wish you a happy, healthy and prosperous New Year 2023
for your comments [ Extensive editing of English language and style required]
you find attachment a new version pf my paper after editing of English language by the center of journal and I made and edit all requirements.

Round 3
Reviewer 1 Report
most of suggestions have been taken into account
Author Response
Thank you.
Reviewer 2 Report
this version was duly revised. I presume editors could consider this manuscript for possible publication subject to English language careful proof reading.
Author Response
English language has been carefully proofread.